# Predictive Factors on the Incidence of Heart Failure in Patients with Ischemic Heart Disease: Using a 10-Year Population-Based Korea National Health Insurance Cohort Data

**DOI:** 10.3390/ijerph17228670

**Published:** 2020-11-22

**Authors:** Seon Young Hwang, Kyung Ae Kim, Oh Jong Choi

**Affiliations:** 1School of Nursing, Hanyang University, Seoul 04763, Korea; 2College of Nursing, Kyoungdong University, Wonju 26495, Korea; kimkyungae9806@gmail.com; 3Graduate School, Hanyang University, Seoul, Seoul 04763, Korea; intpublic20@naver.com

**Keywords:** coronary artery disease, myocardial ischemia, heart failure, risk factors

## Abstract

Early risk stratification and preventative strategies are required in patients with ischemic heart disease (IHD) to prevent heart failure (HF). We aimed to investigate the rate of progression to HF and to investigate the factors predicting the development of HF in a population with IHD for 10 years. A descriptive study was conducted using Korea National Health Insurance Service-National Sample Cohort (NHI-NSC) data (2005–2015). Among the patients diagnosed with IHD for the first time in 2005–2006, 2271 men and 2037 women who responded to the health check-up survey were finally selected. Cox Proportional Hazard regression analyses and the Kaplan–Meier survival analysis were used. HF incidence rates were 5.1% in men and 8.0% in women. The mean duration of transition to HF was 4.85 ± 2.73 years in men and 4.73 ± 2.73 years in women. The non-incidence rate of HF was higher in men than in women (Log-rank test, *p* = 0.0003). Bivariate analyses showed that older age, prevalence of hypertension and diabetes, less alcohol, and lower physical exercise were associated with the incidence of HF in both men and women. Multivariate analyses found that HF incidence in aged subjects ≥70 years was 1.46 times higher in men and 1.44 times higher in women compared to those in their 30 s (*p* < 0.001). Prevalence of hypertension reduced the incidence of HF by 0.78 and 0.87 for men and women, respectively. The prevalence of diabetes increased 1.23 times only in men. These findings suggest that special attention such as periodic counseling and education is needed to prevent progression to HF in elderly and diabetic patients during follow-up after IHD.

## 1. Introduction 

Heart failure (HF) is a critical healthcare issue worldwide that threatens the health of the public due to its high mortality, morbidity, and healthcare costs, including indirect costs [1,2]. The number of patients with HF is increasing as a consequence of aging of the population and high survival rate with improved treatment after cardiovascular disease caused by various risk factors [3]. Although the outcomes have improved with drugs and device therapies, it is known that the number of hospitalizations due to HF increases by more than 1 million every year, and high post-discharge mortality and readmission rates have not changed in the last two decades [4]. In addition, patients with HF suffer from symptoms that cause the patient to visit the hospital, as well as economic loss, and furthermore social psychological difficulties and quality of life is decreased [3,5]. In Korea, the prevalence of HF was estimated to be 1.53%, 12.4 persons per 1000 adults in 2013, with a high socioeconomic burden. It is expected to increase by 2-fold, from 1.60% in 2015 to 3.35% in 2040, indicating that more than 1.7 million Koreans are expected to have HF [6].

Ischemic heart disease (IHD) such as myocardial infarction, which is closely related to individual lifestyle, is the most common cause of developing HF worldwide [2,7]. A prospective cohort study for IHD patients found that a greater number of occluded vessels was an independent risk factor for the occurrence of recurrent myocardial infarction and an indicator of post-MI HF [7]. Of the patients with HF in Korea, 45.4% had IHD, 43.6% had hypertension, and 49.1% had diabetes in 2013 [6]. A Korea multi-center registry for 5625 hospitalized patients with acute HF showed that 37.6% of aggravating factors for acute HF was IHD [8]. Although there are a variety of contributing factors to HF after myocardial infarction such as cardiac dysfunction, inflammation, and remodeling, early risk stratification and preventative strategies need to be defined and required in patients with IHD [2].

These previous studies underscore the need to investigate the processes occurring in the transition from myocardial injury to HF. High-risk patients who will progress to HF after IHD treatment should be identified to provide counseling for medication and lifestyle improvement. Therefore, it is necessary to investigate the rate of progression to HF through long-term follow-up in a population with subsequent IHD after acute phase treatment and to confirm the demographic, clinical and lifestyle factors affecting the development of HF using a cohort study design. We aimed to investigate the factors affecting the development of HF in a population with IHD for 10 years using NHI-NSC data.

To examine the incidence rate of HF among patients with IHD by gender for 10 years.To examine duration of transition to HF from IHD patients.To determine predicting factors affecting the development of HF after the treatment of IHD by gender.

## 2. Materials and Methods

### 2.1. Research Design

This study is a secondary analysis using Korea National Health Insurance Service-National Sample Cohort (NHI-NSC) data with a longitudinal cohort design for 10 years. The NHIS-NSC database is a population-based sample cohort.

### 2.2. Study Participants and Data Collection

The study population was extracted from the data of the NHI-NSC, which is a population-based sample cohort for providing representative, useful health insurance and health examination data. It includes national health checkup materials, medical history, a self-reporting questionnaire and health insurance eligibility information (Qualification DB) from 2005 to 2015 [9]. From the target population, a representative sample cohort of 1,025,340 participants was randomly selected, comprising 2.2% of the total eligible Korean population in 2002, and followed for 11 years until 2013 unless participants’ disqualified eligibility. All Koreans are under the national health insurance system and the National Health Insurance System in Korea is a social security system in which people have been obliged to join since 1989 [10]. The NHI-NSC data included the entire medical records and admission-outpatient data from hospitals, pharmacy and public health. National health checkup data is a governmental regular health screening that is carried out every two years [10]. In health checkups, the main inspection and paperwork response processes are carried out by standardized procedures [11]. Anthropometric measurements were conducted with the same protocols based on the health screening implementation standards provided by Ministry of Health and Welfare [11]. The eligibility information (Qualification DB) of National Health Insurance have information of the demographic and socio-economic status, such as individual’s gender, age, residence area, subscriber type (self-employed insured, insured employees and dependents), birth and death data [9,11]. Inclusion criteria of study subjects were the patients who were newly admitted with IHD in 2005 and 2006 with the exception of those who had previously been admitted with IHD. The disease codes of IHD and HF were identified with the international classification of disease (ICD) 10th edition code I20~I25, respectively. IHD was regarded in this study as unstable angina or myocardial infarction, other ischemic heart disease, or atherosclerotic heart disease. In this study, the diagnosis of IHD and HF was based on the diagnosis code given by the doctor. From 945,226 adults ≥ 30 years old who were treated at a medical institution for two years (2005 and 2006), we selected a total of 11,173 men and women who were diagnosed with IHD for the first time between 2005 and 2006 as a baseline target population. We excluded 329 men and women who were diagnosed with HF before 2005–2006. In addition, non-respondents of health check-ups (*n* = 6527) and those who has no follow-up data including death (*n* = 9) were excluded and then finally 2271 men and 2037 women were selected for this study (Figure 1).

We selected subjects by screening the medical records using the National Health Check-up Database in 2002. This included biochemical data such as cholesterol and glucose, and lifestyle-related information (activity, smoking, alcohol drinking, diet). We utilized an income decile as a socioeconomic indicator of the subject. The income decile is based on the health insurance premium according to income level. In terms of ethical consideration, we did not include the resident registration number because of the protection of personal information [11]. Instead, the data with a personal identification number assigned by National Health Insurance Service was used as a unique number in the research analysis [11].

### 2.3. Ethical Consideration

This study was conducted after approval of the deliberation exemption from the Institutional Review Board of University (IRB# HYI-17-157-1). To protect the patients’ information and identity, all patients received an anonymous identification code in the sample data. The authors could not identify any patients included in the sample data. 

### 2.4. Statistical Analysis

The SPSS/WIN version 24.0 (SPSS Inc., Chicago, IL, USA) was used to verify the basic assumptions of the parametric statistics for the variables. For descriptive statistics, bivariate analyses using χ^2^-tests were conducted to identify the differences of subject characteristics by HF incidence. Cox proportional hazard regression analyses were performed for analysis of predicting factors associated with the incidence of HF in each of the male and female subjects. Kaplan–Meier survival analysis with log-rank tests was performed to examine the differences of incidence of HF in male and female subjects. The entry criterion for the multivariate analysis was *p* < 0.05, which was considered statistically significant.

## 3. Results

### 3.1. Demographic and Clinical Characteristics of the Subjects

General characteristics of the study population are presented in Table 1.

The subjects of this study were 4308 people consisting of 2271 men (52.7%) and 2037 women (47.4%), who were diagnosed with IHD for the first time in 2005 and 2006. The incidence rate of developing HF during follow-up for 10 years was 5.1% (*n* = 115) in men and 8.0% (*n* = 162) in women. The mean duration of transition to HF in IHD patients was 4.85 ± 2.73 years in men and 4.73 ± 2.73 years in women. Women had a significantly shorter duration of transition to HF than men (*p* < 0.001) (Figure 2).

Kaplan–Meier survival analysis showed that the cumulative survival rate was significantly lower in women than men, and the non-incidence rate of HF was higher in men (94.94%) than in women (92.05%) (Log-rank test, *p* = 0.0003) (Figure 3).

### 3.2. Predictive Factors for the Incidence of HF in Men and Women

Bivariate analyses showed that, in men, the incidence of HF was significantly higher with older age, lower BMI, alcohol consumption, hypertension and diabetes (*p* < 0.05). In women, there were significant differences in hypertension, diabetes, and physical exercise (*p* < 0.05) (Table 1).

Cox-Hazard multiple regression analyses found that HF incidence in aged subjects older than 70 years was 1.46 times higher in men and 1.44 times higher in women compared to those in their 30 s (*p* < 0.001). In men and women with hypertension, the incidence of HF was 22% (Hazard Ratio = 0.78, *p* < 0.001) and 13% (Hazard Ratio = 0.87, *p* = 0.025) lower, respectively. However, the prevalence of diabetes increased the incidence of HF 1.23 times only in men (Hazard Ratio = 1.23, *p* = 0.021) (Table 2).

## 4. Discussion

The incidence of HF was higher in females (7.9%) than in males (5.1%), and the incidence of HF increased with higher age in this study. Although it is natural for women to live longer, this finding suggests that care is needed for elderly women with previously treated IHD to enhance health compliance to prevent the progression of HF. As a result of Cox proportional hazard regression analysis, both men and women were 1.46 times and 1.44 times more likely to develop HF in those over 70 years of age than those in their 30s. A review study suggests that this is because aging can lead to limited cardiac regeneration capacity due to excessive oxidative stress and chronic low-grade inflammation [12]. In addition, it is similar to a study on the prognosis of survived patients after myocardial infarction, which showed that older age was a risk stratification tool and independent predictor of 1-year all-cause death or myocardial infarction [13], and an observational study that showed age was a strong predictor of late onset HF following myocardial infarction [14]. The cardiovascular aging process contributes to the development of HF by a complex interaction with risk factors such as obesity, hypertension, atherosclerosis, and accompanying diseases such as diabetes or chronic kidney disease [12]. Therefore, this suggests that if angina or myocardial infarction occurs in the elderly, more efforts are needed to prevent the transition to HF.

In bivariate analyses, diabetes was more likely prevalent in both women and men who developed HF, but in multivariate analysis, it was confirmed as a predictor only in men, where men with diabetes showed 1.23 times more chance to develop HF than men without diabetes. This finding is consistent with a 20 year cohort study for the multivariate hazard ratio for incidence of HF that, when adjusted for age, body mass index, hypertension, high cholesterol, the incidence of HF was 2.98 times higher in adults with myocardial infarction with diabetes compared to those who had myocardial infarction alone without diabetes [15]. In addition, it was similar to observational studies that diabetes was one of the independent risk predictors of developing 1-year all-cause death or myocardial infarction and late onset HF in survivors after myocardial infarction [13,14,15,16]. These results indicate that a strategy for preventing cardiovascular complications in diabetic patients is needed [17]. In particular, further studies on the degree of perception and self-management of male diabetic patients with HF should be conducted.

Previous epidemiological studies have shown that impaired glucose tolerance, increased serum glucose levels and glycated hemoglobin levels are associated with prevalence of systolic HF, as well as prevalence of diastolic dysfunction. HF (28%) and left ventricular dysfunction (23%) were highly prevalent in type 2 diabetes patients [18] and a prospective cohort study with 14.1 years follow-up found that elevated HemoglobinA1c (≥5.5–6.0%) was associated with the incidence of HF among 11,000 adults with no diabetes or HF at baseline [19]. In addition, diabetes accompanied with HF can potentially lead to worse outcomes, with a 25% readmission rate within 6 months and 30% mortality rate within 1 year compared to patients without diabetes [20]. In patients with advanced diabetes, it is often accompanied by hypertension, microvascular dysfunction, kidney damage and accelerated cardiac dysfunction [20]. In particular, adults with diabetes have a high risk of HF but often have fewer symptoms of HF and no structural heart disease [21]. Therefore, adults with diabetes, especially post-myocardial infarction men with diabetes, should pay attention to monitoring blood sugar levels, and modifying and managing lifestyle risk factors as soon as possible according to guidelines to prevent progression to HF [20,22]. Conversely, a cohort study of 50,874 HF patients in Europe showed that HF severity predicted the risk of developing diabetes after myocardial infarction [23]. This dual nature of HF and diabetes highlights the importance of preventing HF in diabetic patients and early diagnosis of diabetes in HF subjects.

In this current study, patients with a history of hypertension had 17.2% lower incidence of HF than those with no hypertension. This finding is similar to a prospective study of acute HF patients in Korea where univariate analysis showed that patients who were prescribed beta-blockers, angiotensin converting enzyme inhibitors (ACEI), or angiotensin II receptor blockers (ARB) at discharge had relatively low hospitalization and mortality rates within 30 days [24]. This is because the antihypertensive agents have a similar pharmacological action to the beta blockers and ACEI or ARB. Hypertension is the one of the most prevalent diseases in Korea and its prevalence is about three times higher than diabetes. Compared to diabetes, it is found at an early age and the antihypertensive drugs work better, so if you take the drug well, blood pressure is well controlled. Therefore, compliance with medication should be recommended in patients with IHD who have hypertension to prevent transition to HF.

In this study, we examined the relationships between lifestyle factors and the prevalence of HF in IHD patients. Bivariate analyses showed that male patients who drank no alcohol or drank less alcohol were more likely to develop HF in this study. This is consistent with a review study that reported that small to moderate alcohol drinking has a beneficial effect on overall cardiovascular profile and mortality [25], but it was not supported by the multivariate analysis in this study. Thus, further robust studies including a randomized controlled study need to be conducted to determine a strong causality of its protective effect.

In addition, male and female patients who developed HF were less likely to do physical exercise compared to those with no HF in the bivariate analysis of this study. However, a significant relationship was not supported in a multivariate analysis, which is similar to a cohort study, where Cox-Hazard regression analysis showed no association between leisure time physical activity and the risk of HF or all-cause mortality after myocardial infarction [26]. In the current study, only frequency of exercise, more than one time per week, was used as a categorical variable. Therefore, further study will be needed with qualitative considerations such as exercise type and total daily amount of physical activity to examine the relationship between low physical activity and developing HF. 

This study has the following limitations. The original study subjects were 10,844 subjects who were screened in 2005 and 2006, but we tried to combine them with health examination data to identify lifestyle factors. Finally, as a possible combination, the number of subjects in this study was reduced to 4317. In addition, the health examination survey on the daily life habits of smoking, alcohol, and drinking was conducted by using a self-reported questionnaire, where there was a limit to accurate analysis.

## 5. Conclusions

These findings suggest that special attention is needed to prevent progression to HF in the elderly over 70 years old and diabetic patients who are managed after IHD diagnosis. IHD patients visiting the outpatient clinic, especially the elderly and diabetic, should be provided with periodic counseling and education to help them manage their comorbidities, medications and self-care, and to recognize HF symptoms early.

## Figures and Tables

**Figure 1 ijerph-17-08670-f001:**
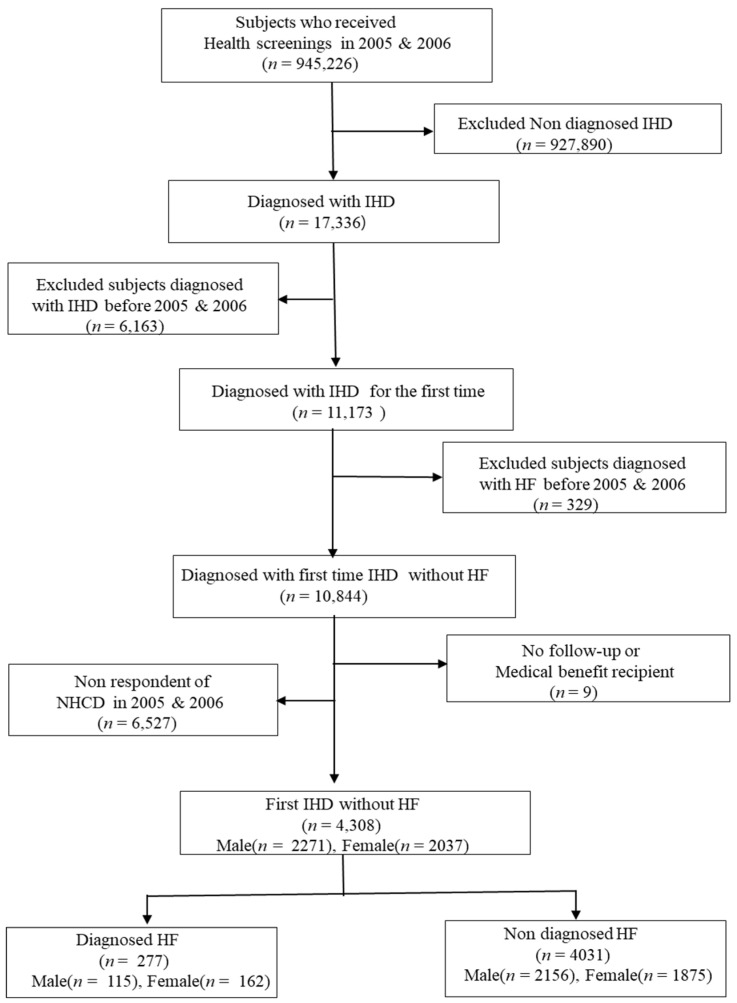
Flow chart of the study population. IHD: ischemic heart disease, NHCD: national health check-up data, HF: heart failure.

**Figure 2 ijerph-17-08670-f002:**
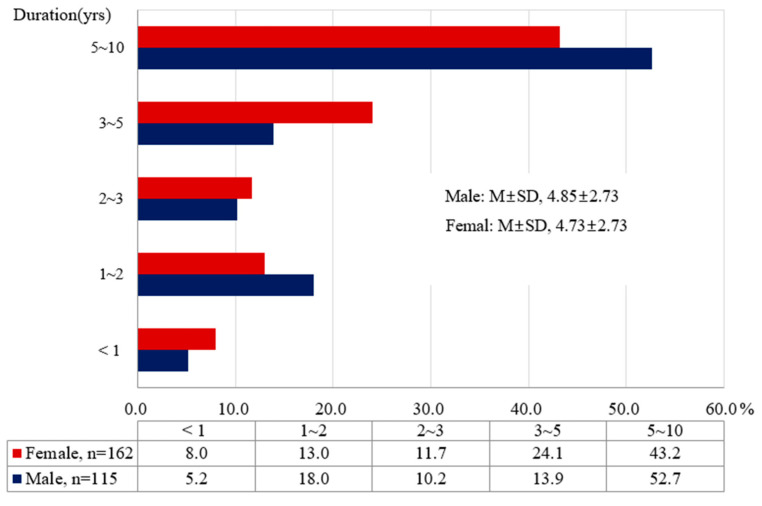
Duration of developing heart failure after ischemic heart disease.

**Figure 3 ijerph-17-08670-f003:**
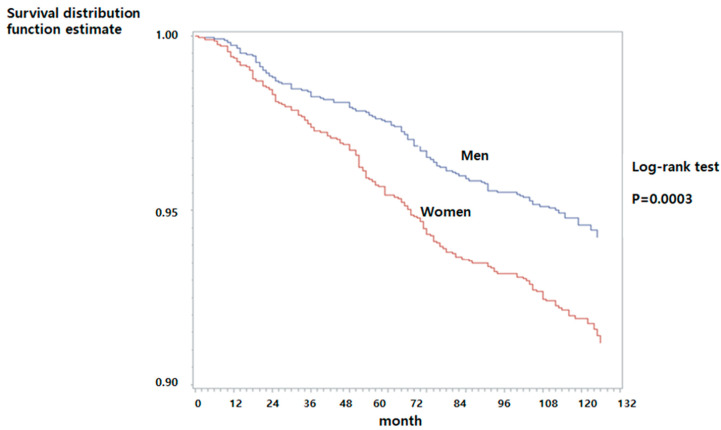
Survival curve for heart failure occurrence according to gender.

**Table 1 ijerph-17-08670-t001:** Relationship between subject’s characteristics and incidence of HF by gender.

Variables	Men (*n* = 2271)		*p*	Women (*n* = 2037)		*p*
	No HF (*n* = 2156, %)	HF (*n* = 115, %)		No HF (*n* = 1875, %)	HF (*n* = 162, %)	
Age, yr M ± SD	54.1 ± 12.7		<0.001	58.4 ± 11.1		<0.001
30–39	306 (14.2)	1 (0.8)		83 (4.4)	0 (0.0)	
40–49	532 (24.7)	11 (9.7)		344 (18.4)	5 (3.1)	
50–59	585 (27.1)	30 (26.0)		593 (31.6)	33 (20.4)	
60–69	494 (22.9)	43 (37.4)		585 (31.2)	64 (39.5)	
70–85	239 (11.1)	30 (26.1)		270 (14.4)	60 (37.0)	
Household income * (*M* = 2235, *F* = 2014)			0.912			0.594
Low (1–2 decile)	242 (11.4)	14 (12.2)		324 (17.5)	32 (19.7)	
Middle (3–8 decile)	1113 (52.5)	62 (53.9)		929 (50.2)	74 (45.7)	
High (9–10 decile)	766 (36.1)	39 (33.9)		600 (32.3)	56 (34.6)	
Body Mass Index			0.021			0.119
Normal < 23 Kg/m^2^	626 (29.0)	47 (40.9)		637 (34.0)	45 (27.7)	
Overweight < 23–25 Kg/m^2^	577 (26.8)	23 (20.0)		532 (28.4)	44 (27.2)	
Obesity ≥ 25 Kg/m^2^	953 (44.2)	45 (39.1)		705 (37.6)	73 (45.1)	
Hypertension, yes	388 (20.0)	31 (27.0)	0.016	467 (24.9)	63 (38.9)	<0.001
Diabetes, yes	142 (6.6)	19 (16.5)	<0.001	126 (6.7)	27 (16.7)	<0.001
Family history of stroke, yes	21 (5.6)	4 (3.5)	0.336	116 (6.2)	7 (4.3)	0.348
Total cholesterol ≥ 240 mg/dL	243 (11.3)	17 (14.8)	0.253	309 (16.5)	29 (17.9)	0.618
Alcohol drinking (*M* = 2224, *F* = 1978)			0.027			0.228
No or Seldom	933 (44.2)	63 (56.3)		1585 (87.0)	143 (91.7)	
Moderate (1–2 times/week)	854 (40.4)	32 (28.6)		210 (11.5)	12 (7.7)	
Severe (>3 times/week)	325 (15.4)	17 (15.1)		27 (1.5)	1 (0.6)	
Smoking *(M* = 2213, *F* = 1952)			0.733			0.624
Never	1081 (51.4)	61 (54.4)		1734 (96.5)	148 (95.5)	
Ex-smoker	365 (17.4)	20 (17.8)		16 (8.9)	1 (0.6)	
Current smoker	655 (31.2)	31 (27.8)		47 (2.6)	6 (3.9)	
Exercise *(M* = 2197, *F* = 1965)			0.009			0.006
Seldom (No or hardly ever)	985 (47.2)	66 (60.0)		1098 (60.6)	110 (71.9)	
Regular (≥1 time/ week)	1102 (52.8)	44 (40.0)		714 (39.4)	43 (28.1)	

* The income decile is based on the health insurance premium according to income level, indicating that the higher the level, the richer. HF: heart failure.

**Table 2 ijerph-17-08670-t002:** Cox-Hazard regression analysis for predicting factors on incidence of heart failure by gender.

Variables	Male (*n* = 2271) No HF (*n* = 2156) vs. HF (*n* = 115)					Female (*n* = 2037) No HF (*n* = 1915) vs. HF (*n* = 162)				
	β	SE	*p*	HR	95% CI	β	SE	*p*	HR	95% CI
Age, yr (ref: 30–39)										
40–49	−0.04	0.07	0.587	0.96	0.85–1.10	0.25	0.13	0.054	1.29	0.95–0.99
50–59	−0.04	0.07	0.594	0.96	0.91–1.13	0.18	0.13	0.154	1.20	0.82–0.93
60–69	0.06	0.08	0.404	1.07	0.68–1.05	0.17	0.13	0.202	1.18	0.75–0.81
70–85	0.38	0.09	<0.001	1.46	1.31–1.55	0.36	0.14	<0.001	1.44	1.05–1.45
Body Mass Index, kg/m^2^ (ref: <23)										
23–25	−0.03	0.06	0.601	0.97	0.91–1.35	−0.11	0.06	0.085	0.90	0.91–0.98
≥25	−0.02	0.05	0.763	0.98	0.83–1.05	−0.09	0.06	0.151	0.92	0.68–0.82
Hypertension (ref: No)										
Yes	−0.24	0.06	<0.001	0.78	1.15–1.48	−0.14	0.06	0.025	0.87	1.09–1.78
Diabetes (ref: No)										
Yes	0.21	0.09	0.021	1.23	1.06–1.35	0.08	0.10	0.380	1.09	0.78–0.85
Exercise (ref: Regular)										
Seldom	−0.04	0.04	0.302	0.95	0.80–1.15	−0.08	0.05	0.102	0.92	0.68–0.88

SE: standard error; HR: hazard ratio.

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
