# Peer review of "Predictive Factors on the Incidence of Heart Failure in Patients with Ischemic Heart Disease: Using a 10-Year Population-Based Korea National Health Insurance Cohort Data"

_ijerph, 2020, doi:10.3390/ijerph17228670_

Round 1

Reviewer 1 Report

The work is methodologically correct except for a series of issues that the authors must take into account: The research work was carried out 5 years ago, most of the citations used in the bibliography are more recent than the work itself. How can you compare an older study with authors who have recently done their research? The inclusion and exclusion criteria are confusing, in figure 1, it establishes two exactly the same exclusions, they should explain the difference. The quality of figure 3 needs to be improved. The discussion does not add anything new that authors with more recent works on the same population have already described. A reiterative conclusion, which says nothing new.

Reviewer 2 Report

Generally speaking, this paper is okay and appropriate to a certain extent, but it is not innovative enough. The analysis is simple and not perfect, which needs further modification.

Methods:

1. The baseline characteristics of the subjects of the NHI-NSC cohort need to briefly introduce.

2. Why did the author select people over 30 years old for the study? Besides, the diagnostic criteria for IHD and HF need to be supplemented.

Results:

3. It is recommended that the author add a description of the death outcome and supplement the average follow-up time of the cohort of this study.

4. As for cox regression analysis, the result is too simple. First, why didn't the author analyze the total population first? And how did the author determine which independent variables to include in the Cox regression model?

5. As is known to all, there are many risk factors for HF, such as hypertension. It is recommended to do further sensitivity analysis to verify the results of this study from multiple perspectives.

6. There are some small issues in format that need attention, such as the age variable, the 95% confidence interval of HR should be completed in table 2, etc.

Discussion:

7. It is suggested to revise the discussion, and the accumulation of literature is not sufficient, and further revision is needed.

8. The study suggests that the incidence of HF in women was higher than that in men, but did not statistically test whether this difference is meaningful.

9. Regarding age, according to the results part, HR value only increased meaningfully in the 70+ age group, so this statement needs to be reconsidered.

10. Part of the description in the discussion part was aimed at the whole population, so it is recommended to add the relevant indicator descriptions of the population in the results part.

Reviewer 3 Report

The authors examined the clinical factors predicting new-onset heart failure and incidence using national insurance data. They found that older age, hypertension and diabetes, less alcohol drinking, lower physical exercise were associated with the incident HF in both men and women. This paper is well planned and well written. I have some minor comments.

The results are not much different from previous reports and lack new information. If you have data on 12-lead ECG, revascularization, residual ischemia, and cardiac function, please reanalyze the Cox model.

Please discuss the incidence rate of heart failure in this study with previous data.

Round 2

Reviewer 1 Report

The changes and clarifications requested have been made by the authors.

Reviewer 2 Report

The content of the manuscript is complete and the result is reasonable. The author carefully revised the manuscript based on the comments. I have no further comments or questions.